:ᐸ᛫᛫ PLOS | ONE

# Towards a global understanding of the drivers of marine and terrestrial biodiversity

Tyler O. Gagné[1], Gabriel Reygondeau[2], Clinton N. Jenkins[3], Joseph O. Sexton[4], Steven J. Bograd[5], Elliott L. Hazen[5], Kyle S. Van Houtan[1,6]*

**1** Monterey Bay Aquarium, Monterey, CA, United States of America, **2** Nippon Foundation, Nereus Program and Changing Ocean Research Unit, Institute for the Oceans and Fisheries, The University of British Columbia, Vancouver, British Columbia, Canada, **3** IPÊ—Instituto de Pesquisas Ecológicas, Nazaré Paulista, São Paulo, Brazil, **4** terraPulse, Inc., North Potomac, Rockville, MD, United States of America, **5** NOAA, Environmental Research Division, Southwest Fisheries Science Center, Monterey, CA, United States of America, **6** Nicholas School of the Environment, Duke University, Durham, NC, United States of America

☯ These authors contributed equally to this work.
* kvanhoutan@mbayaq.org

## Abstract

Understanding the distribution of life's variety has driven naturalists and scientists for centuries, yet this has been constrained both by the available data and the models needed for their analysis. Here we compiled data for over 67,000 marine and terrestrial species and used artificial neural networks to model species richness with the state and variability of climate, productivity, and multiple other environmental variables. We find terrestrial diversity is better predicted by the available environmental drivers than is marine diversity, and that marine diversity can be predicted with a smaller set of variables. Ecological mechanisms such as geographic isolation and structural complexity appear to explain model residuals and also identify regions and processes that deserve further attention at the global scale. Improving estimates of the relationships between the patterns of global biodiversity, and the environmental mechanisms that support them, should help in efforts to mitigate the impacts of climate change and provide guidance for adapting to life in the Anthropocene.

**Data Availability Statement:** All relevant data are available from the Open Science Framework at https://osf.io/jm7fn/.

## Introduction

Empirical and theoretical approaches to understanding biodiversity have historically focused on particular taxonomic groups, geographic domains, and explanatory variables and therefore have not evaluated relationships at a truly global scale. Studies concentrating on birds and mammals, on terrestrial or marine species, and on individual mechanisms such as latitude [1–5] have produced critical knowledge that has advanced ecology and conservation. Today the unprecedented availability of biological and environmental data, as well as machine learning models useful for large and complex datasets, however, provide new opportunities to answer questions about biodiversity. Aside from comparing and contrasting patterns across domains of data, there is a greater chance to resolve the effects [6] of specific drivers [7, 8] on species richness in a more robust way, and to describe the interaction and gradient [9] of multiple

**Funding:** This work was supported by a Presidential Early Career Award for Scientists and Engineers (PECASE) to KV and by generous contributions to the Monterey Bay Aquarium a non-profit, 501(c)(3) tax-exempt organization. terraPulse Inc., an incorporated research services company, provided support in the form of salaries for JS, but did not have any additional role in the study design, data collection and analysis, decision to publish, or preparation of the manuscript. The specific roles of the authors are articulated in the Author Contributions section.

**Competing interests:** Competing Interests: The authors have read the journal's policies and declare the following competing interest: JS has a commercial affiliation (terraPulse, Inc.) and is a paid employee of terraPulse, Inc. This does not alter our adherence to PLOS ONE policies on sharing data and materials.

drivers and their functional forms. The resulting advances could not be more urgent as the threats from climate change are increasingly destabilizing natural ecosystems, driving extinctions, and subsequently disrupting human socioeconomic frameworks [10–12].

Patterns of species richness are the result of ecological and evolutionary processes acting over geological time scales. Both on land and in the ocean–which we refer to as "domains"–the tropics have notable peaks of biodiversity, presumably because they have housed a stable and favourable confluence of diversity drivers for thousands of years [1, 3, 13, 14]. Of these drivers, temperature and sunlight are considered broadly important, as is primary productivity (i.e., the species-energy hypothesis [15]). However, several domain-specific variables such as dissolved oxygen in the ocean and precipitation on land also uniquely influence the evolution and ecology of species and their distributions in each domain [16, 17]. Improved spatiotemporal resolution of these and a host of other variables are increasingly available, giving a chance to better characterize and assess their variability and importantly their impact on biodiversity. Corresponding advances in biodiversity monitoring [18] means the effect of environmental forcing can be assessed across a more representative swath of taxonomic groups, and no longer limited to the previously well-studied taxa. The substantial progress in both of these areas can help to develop a fuller picture of the processes that drive species richness, which is in turn critical for understanding how these patterns may be impacted by climate change [19].

In this study, we assembled data on the distributions of 44,575 marine species (10,873 fishes, 9,582 macroinvertebrates, 7,663 arthropods, 5,753 microinvertebrates, 3,976 molluscs, 2,780 cnidarians, 2,580 worms, 1,175 echinoderms, 126 mammals, 67 reptiles,) and 22,830 terrestrial species (10,959 birds, 6,407 amphibians, 5,464 mammals). To constrain comparisons across domains as much as possible, we conducted the formal analysis both with and without marine invertebrates (see supporting information). We also assembled a suite of likely predictive environmental variables. These variables characterized the central tendency, variation, and seasonality of abiotic and biotic drivers including primary production, temperature, solar energy, biogeochemical resources, and the physical environment. As we hope to understand the influence of primary production [20, 21] on species richness patterns, we excluded primary producer taxa (e.g. phytoplankton, macroalgae, trees) from the species richness calculation to avoid any circularity in the modelling. Collectively, therefore, we have combined the two largest datasets on biodiversity in the marine and terrestrial domains, recognizing that taxonomic representation differs between land and sea. These taxonomic differences lead to comparable datasets that represent domain-specific biodiversity, even though they may vary when compared in terms of overall composition. We have therefore limited our comparisons to consumers and feel this provides a best-available approach towards understanding drivers of biodiversity on land and in the sea.

Aside from a more expansive approach to data, here we use artificial neural networks (ANNs) to predict species richness in each domain as a function of the prevailing environmental features. The ANN approach is an improvement upon previous modelling methods for three main reasons. The first and most obvious is that ANNs are more accommodating to big data frameworks in terms of computational performance [22]. Here we analyze tens of thousands of species distributions, resolved globally at a spatial resolution of 50x50 km (each dataset containing a potential $\geq$ 200,000 elements), using 30 sets similarly resolved environmental drivers. While our approach may be modest by comparison to more traditional big data analyses, the ANN framework we employ here is uniquely more scalable for when such datasets are eventually available for our specific application here. Secondly, the ANN approach provides new information about the modelled driver relationships using sensitivity analysis of variable importance and variable interactions (known as multivariate partial dependencies). Thirdly, ANNs use a permutative approach to weight the contribution of each variable in order to find the best average or neural pathway to the output or response variable. Therefore, compared to

more traditional modelling approaches like generalized linear models or generalized additive mixed models, ANNs use the data to develop a robust weighting scheme over all the environmental variables used to feed the model. Like ANNs, generalized additive mixed models also use nonlinear splines [23, 24] to model ecological relationships, and therefore are an improvement from linear approaches [25–27]. However, for the above reasons it makes sense to explore ANNs for the added performance of big data scalability and emerging information on driver interactions. Ultimately, combining comprehensive data streams and analytic tools allows us to understand environmental drivers of biodiversity within and across domains and reveal locations that defy expectations based on existing data.

## Material and methods

### Species richness

To train the models in the terrestrial domain we used publicly available species range data. In the marine domain, we used high-volume, screened, and sampling-effort-corrected occurrence data that produced approximately 51 million records post-screen [28]. While the underlying datasets ultimately differ, we used this approach as it represents the best-available data for each domain, it yielded extensive taxonomic representation, and is demonstrated to align with other methods of estimating marine species richness [29]. By using this approach, we are able to maximize the available taxa used for each domain, and characterize broad-scale species richness patterns with the most data possible, at a relatively high resolution of 50x50 km [28–30] for all environmental variables in the model.

For the terrestrial domain, we compiled species ranges using published methods and sources [1]. Distribution data of birds ($n = 10959$), amphibians ($n = 6407$), and mammals ($n = 5464$) were from the International Union for the Conservation of the Nature [31] and BirdLife International [32], and are expert-reviewed and rigorously quality controlled [1]. These overlaid polygon range maps are drawn from expert consensus, and are the best-available data, at the global scale, for terrestrial biodiversity. We excluded polygons of all extinct or non-native ranges (invasions and introductions arising from anthropogenic activity), as well as seabird species. The final terrestrial richness layer consisted of 22,830 species.

For the marine domain, we queried marine richness data from the Ocean Biogeographic Information System, the Global Biodiversity Information Facility, FishBase, the Jellyfish Database Initiative, and the International Union for the Conservation of the Nature. From the ensemble (>1 billion entries), we performed several quality control routines. We first cleaned records of spatial NULL values, removed records with no definition to species level, expunged duplicates, and assigned full (updated when necessary using the World Register of Marine Species [33]) taxonomic information using the Taxize library [34]. Additional documentation and justification for the vetting methodology used for marine records is described in detail elsewhere [18, 30, 35–37]. Briefly here, we screened point observation occurrences and removed extremely implausible values based on the ratio of the number of independent records in time and space relative to the latitudinal and thermal range of the species [36, 37]. For each species, a random ($1 < n < 1000$) number of records was selected and the thermal and latitudinal range estimated. This was repeated 1000 times. We then confronted the simulated latitudinal range and thermal range (1,000 simulations) to values obtained using all the occurrence records gathered on the species. We computed a confidence interval of the known range by quantifying the difference between the 1st and 99th percentile of observed latitude coordinates and thermal value, assuming that the acceptable number of records to capture the latitudinal and thermal range was obtained when more than 950 randomly selected records were comprised within the confidence interval determined from the global records. The median number of

points per species found to capture this computed latitudinal range was 33 (+/- 4) records, and 41 (+/- 3) records for computed thermal range. Species with less than 41 independent recorded observations were removed from further analysis. We then checked for the potential influence of systematic sampling bias by developing rarefaction curves [38] across latitudinal bins (see S16 Fig).

The final marine biodiversity dataset comprised taxonomic information and filtered occurrences for 44,575 species (10,873 fish, 9,582 macroinvertebrates, 7,663 arthropods, 5,753 microinvertebrates, 3,976 molluscs, 2,780 cnidarians, 2,580 worms, 1,175 echinoderms, 126 mammals, 67 reptiles) for a total of 51,459,235 records. While this represents the largest spatially-explicit dataset of marine species ever described [3, 35, 39], it remains a fraction (~18%) of the named marine species [29, 33].

All species richness was ln(x+1) transformed and all data projected to a cylindrical equal area 50 km × 50 km grid. The script for the R Code is provided in the online repository.

**Considering the impact of marine invertebrate taxa on model interpretation.**   It is possible that the varied patterns between marine and terrestrial domains may emerge simply as a result of including invertebrate taxa in one realm and not including it in the other (due to data availability alone). To address this possibility, we ran an ancillary analysis and have supplied the outputs in the supplementary material (see S10–S15 Figs). The comparisons presented in the main text by including all marine species (with invertebrates) showed minor differences, and no broad changes, to the results when invertebrates are excluded. Nonetheless, we provide the additional analysis for comparison and to encourage and further exploration.

## Environmental drivers

We gathered a suite of 21 globally distributed environmental datasets, spanning terrestrial and marine domains, with an additional 18 data series representing domain-specific drivers. This provided a total of 30 environmental driver input variables for the terrestrial and the marine ANN. S1 Table and S1–S3 Figs provide more details on the series and their spatial and statistical distributions.

**Normalized Difference Vegetation Index (NDVI) and chlorophyll-A (Chl-A).**   We retrieved monthly means of NDVI and near-surface concentration of chlorophyll-A from NASA Earth Observation servers from 2003–2017. This imagery was resampled, assembled, and re-projected from georeferenced sinusoidal tile images gathered by the Moderate Resolution Imaging Spectroradiometer (MODIS) sensor. NDVI is the MOD13A2 product and chlorophyll-A is MODIS-Aqua Level-3 Binned Chlorophyll Data Version 2014. We refer to vegetation indices as proxies for net primary production throughout the manuscript [40, 41]. We did not use NPP products as methods for derivation are debated [20, 42]; however, this framework is flexible enough to incorporate any NPP products in the future.

**Elevation and depth.**   Terrestrial elevation data were from the 2015 release of the Shuttle Radar Topography Mission [43]. Ocean depth is the General Bathymetry Chart of the Ocean, a global 30-arc second interval grid [44]. The General Bathymetry Chart of the Ocean grid is a continuous terrain model for the ocean that combines quality-controlled depth soundings with interpolation between sounding points informed by satellite derived gravity measurements. At a global scale, the marine driver data series (e.g., production, sunlight, oxygen) are provided and only available from sensors of the ocean surface. Though these environmental variables vary continuously with depth, at this time, we cannot include this as a model factor. Therefore, the variable of depth may serve as a proxy until such data are resolved globally.

**Temperature, precipitation, and dissolved oxygen.**   Temperature time series (2007 to 2017) are from the European Centre for Medium Range Forecasts. The SST and SAT products

are a reanalysis of temperature based on satellite sensing and station measurement [45]. Precipitation data is Global Precipitation Climatology Centre 0.5-degree dataset. Measurements are interpolated gauge based precipitation totals on a monthly time step from 2007 to 2017 from ~7,000 stations [46]. Dissolved oxygen is the integrated 30m data from the 2013 World Ocean Atlas. The World Ocean Atlas is a 1-degree grid developed from *in situ* interpolated surface measurements on a monthly time step.

**Solar insolation.** We obtained monthly insolation for 2006 to 2017. The Clouds and the Earth's Radiant Energy System sensor approximates surface solar insolation by measuring escaping radiant energy while accounting for attenuation due to atmospheric conditions, angle of incidence, and slope aspect [47].

## Feature engineering

We developed a selection of features to describe the inter- and intra-annual central tendency, variance, and seasonal phenology of the driver variables. Our dataset was composed of 29 individual inputs. We used arithmetic mean to describe central tendency, while we described variability with standard deviation, range, and coefficient of variation. Upper tails of annual sums of NDVI and Chl-A were truncated to 99% quantiles, and all features were truncated to 1% and 99% quantile values. Elevation and depth data were input as described, without quantile truncation. S1 Table provides further details on all of the input feature data.

Seasonal patterns of primary production can be important to the dynamics of food webs. We utilized continuous wavelet transformation metrics to describe the phenology of primary production and solar energy. This approach draws from earlier work on phytoplankton production [18]. We describe 12-month and 6-month seasonal phenology intensity by calculating the mean of the wavelet power spectrum of primary production and solar insolation. This process was conducted for each cell in a raster through time, with the final estimate being a measure of 6-month and 12-month seasonal intensity. S5 Fig provides more details.

To facilitate training the ANN, all environmental driver datasets were rescaled 0–1 so that all model inputs were on the same magnitude. While ANNs have the advantage over previous approaches of not having to explicitly define parameter relationships or interactions, some pre-processing of the model inputs is necessary.

## Model development

We developed two discrete feed-forward neural networks to understand species richness in the marine and terrestrial domains. Neural networks have shown incredible power at pattern recognition, parameterizing nonlinearity, and identifying interactions [6]. However, they have generally been regarded as a "black-box" with weak opportunities for model interpretation. We argue against this and present several methods to better analyze the model's interpretation of the input-output relationship, which in our case here is the effect of environmental drivers on species richness. Our approaches here include mapping the spatial distribution of model residuals (Fig 1), approximating variable importance through perturbation and resampling (Fig 2), and plotting multivariate partial dependency (Fig 3). This helps both to understand where the models performed well, where opportunities for improvement lie, all couched within nonlinear and multivariate (in our case 30 variables) models whose main interpretive results (Figs 2 and 3) were attained with sensitivity analyses (see *Variable importance sensitivity* below).

**Model fitting.** We used the MXNet library within an R package wrapper for the development of the feed forward ANNs. MXNet is open-source ANN framework that is flexible, supports multiple programming environments, and is scalable [48]. We fit the ANNs on an 80%

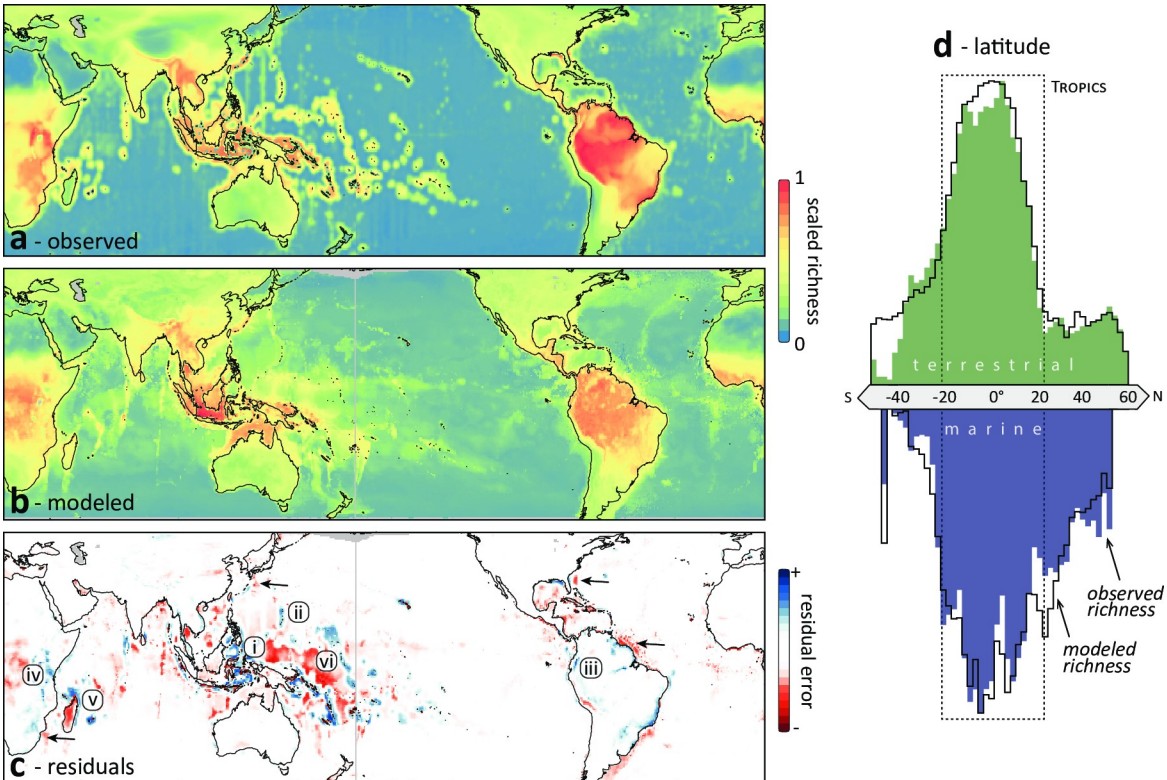

**Fig 1. Global terrestrial and marine biodiversity patterns. (a)** Observed species richness derived from the distributions of 44,575 marine and 22,830 terrestrial species. Species richness is *ln*-transformed and rescaled within each domain (terrestrial and marine) and plotted on a 50 km equal area grid. **(b)** Artificial neural network model predictions (ANNs) of species richness considering a suite of 29 environmental drivers. **(c)** Model residuals highlight areas that are particularly species-rich (underpredicted, blue) and species-poor (overpredicted, red) regions relative to the underlying environmental drivers. These highlight locations of exceptional biodiversity such as reef ecosystems of the (i) Coral Triangle and (ii) Marianas Archipelago and wet forests of the (iii) tropical Andes and (iv) Eastern Arc mountains. It also identifies species-poor settings like isolated islands (v, Madagascar) and major biogeographic boundaries in the ocean (vi, Andesite line). Arrows designate species-poor marine regions with high velocity boundary currents. **(d)** Latitude does not affect model performance, as there are no systematic meridional differences between observed and modelled richness. The northern-hemisphere bias of land, and the corresponding abundance of shallow ocean environments, generates a similar imbalance of marine species richness. Chart area represents the average species richness, zonally, in 2° latitude bins.

training subset of the data and tested performance on a 20% test split. Like more traditional modeling approaches, spurious interactions can be parametrized (but not through user curation) in an ANN, but this tends to occur in situations where a model is trained on a small dataset that is not a representative sample of a broader population. In the case of our dataset, and the chosen parameterization of the ANN, our approach is resilient to spurious interactions being trained in the model. To maximize generalizability, hyperparameters were selected to minimize 5-fold cross validation root mean square error variance on the training set. Each ANN was fitted with 0.2 dropout, rectifier activation for 3 hidden layers with 10 nodes each, 0.9 momentum, a batch size of 128, for 10 epochs.

**Residual mapping.** Fig 1A shows the observed species richness upon which the models were trained. Fig 1B shows the predicted species richness given how the models interpret relationships between the feature inputs and the observed richness. Fig 1C spatially plots the residuals of the observed versus the predicted species richness. Plotting the residuals acknowledges where the model still cannot approximate richness given the features supplied. This difference fosters insight into where species richness is unexpected relative to relationships estimated with environmental variables.

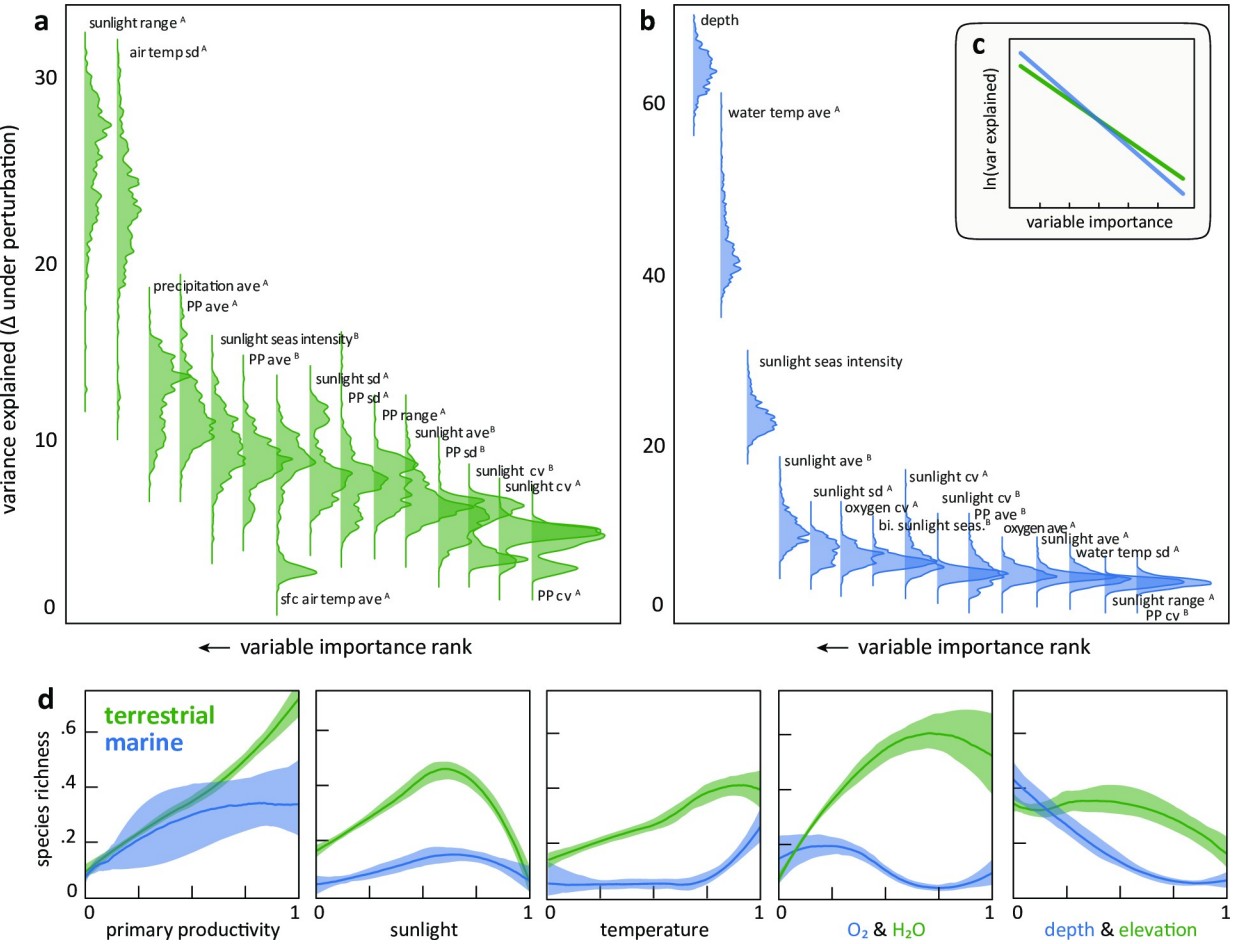

**Fig 2. Environmental drivers of species richness in marine and terrestrial domains.** The ranked importance of the top 15 environmental variables in the (a) terrestrial and (b) marine ANNs. Bootstrapped driver variable importance plots display the densities of change in explained variance made when perturbing individual model factors. The process is repeated 500 times with a random subset of the data on which a new ANN model is trained, generating a robust importance ranking by allowing multiple weight matrices to be evaluated as to how they learn the driver-richness relationship. On the variable labels, "A" is measured within year, "B" is between years. (c) Inset plot shows these same relationships on a log-linear scale, highlighting the steeper decline of variable importance and therefore greater effect of fewer variables in marine systems. (d) Pairwise plots showing the effects from a subset of individual drivers on species richness. To alleviate overplotting, graphs display the median ensemble (with 95% quantiles) of 100 local regression models fit from resamples of the global dataset. For (d) all variables are annual means and rescaled to 0–1 without further transformation. Similar factors are plotted when the same do not exist in both domains. Supporting information provide the full list of all modelled variables and their explanation.

**Resampled pairwise comparison.**  Fig 2D shows the bivariate relationships for a select set of features between domains (full feature set in supplement). We developed these plots by sub-sampling 10000 points from each domain and fitting a loess through the subsample. This was repeated 100 times and we calculated the 95% quantiles and median estimates of the models.

**Variable importance sensitivity.**  We developed a sensitivity analysis to test variable importance and improve insight into the modelled relationships. Fig 2A–2C is a summary of this resampling results to estimate the variance of variable importance. We repeatedly fit ANNs to subsets of the data and tested for model performance decline when individual variables are replaced with resampled noise. With this we can get distributions of variable importance rankings that better approximate the stochastic range of variable importance.

**Multivariate partial dependence plots.**  ANNs can parametrize interactions between variables without having to explicitly define a term in the model. Therefore, we can explore all

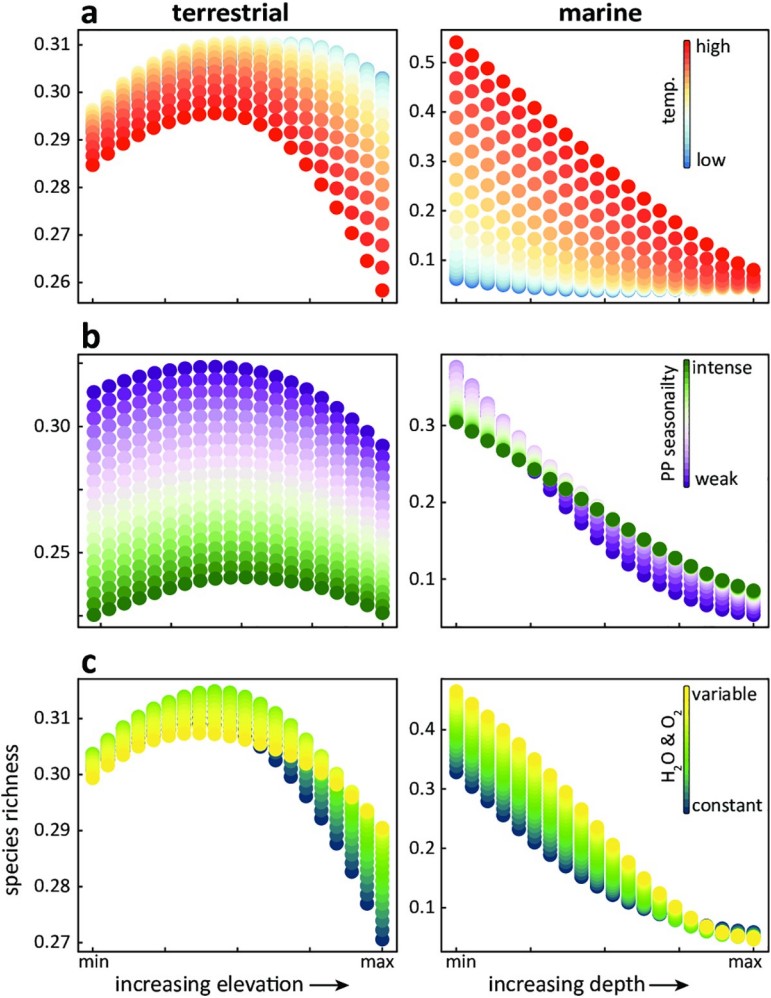

**Fig 3. Multivariate partial dependence plots offer a chance to better grasp driver interactions.** These plots show the neural networks' approximation of driver richness relationships when two inputs co-vary. In this example, we see how species richness is expected to respond across different temperatures (a), seasonality in primary productivity (b) and limiting biogeochemical variables (c) in response to changing depth and elevation. This is an example of the power of ANNs to approximate functions and the integration of interactions. This can be explored for any possible interaction.

potential multivariate relationships the model has approximated between the environmental variables and species richness. Fig 3 is an example of this, where we visualize the relationship between depth/elevation across temperature, primary production seasonality, and limiting biogeochemical variables ($H_2O$ and $O_2$).

**Comment on spatial autocorrelation.** Spatial autocorrelation (SAC) refers to the process by which variation between values in space are affected by the relative spatial distances from one another. It is important though to distinguish between the forms autocorrelation can take [49]. SAC can manifest in the raw values of the response variable and in the values of the residuals of a regression model itself. SAC in raw response data and/or the residuals is often reflective of underlying drivers and latent effects. However, it is the persistence of the appearance of autocorrelation in model residuals that is reflective of bias or distortion in a model with unaccounted for inputs [49]. There are several methods to absorb residual autocorrelation when trying to estimate unbiased model parameters. Most common methods utilize some form of an area weighted auto-covariate in the regression model. While the process of incorporating

these auto-covariates can improve prediction power, they can and often do underrepresent included effects and mask unaccounted for underlying ecological process that may drive residual spatial autocorrelation (RSA). The RSA can reflect unexplained natural process. Therefore, while we did investigate auto-covariate development and RSA quantification via semi-variograms (see code repository), we believe the presentation of model residuals transparently opens discussion around appropriate model inputs. We ultimately present the spatial distribution of the residuals of the model without an RSA covariate to facilitate the identification of important processes that may not be accounted for in the full model, rather than mask those latent processes.

## Data and code repository

All models were run in the R environment [50] with visualizations created using ggplot2 [51] and figures compiled and postprocessed using Adobe Illustrator [52]. All data, as well as markdown files with annotated and commented codes and scripts are available in a third-party open-access repository (https://osf.io/jm7fn) through the Open Science Framework.

## Results and discussion

The global patterns of species richness that we describe (Fig 1A) followed earlier published findings [3, 11]. Given our selection of driver inputs, the ANN had reasonable accuracy in predicting the observed species richness (Fig 1B: Test set: $R^2_{land} = 0.81$, $R^2_{ocean} = 0.69$, $RMSE_{land} = 0.08$, $RMSE_{ocean} = 0.09$).

Examining the difference between the observed and modelled richness, the model residuals, revealed where the environmental drivers do not fully explain the observed patterns (Fig 1C). These outliers, 'bright' and 'dark' spots [53] of diversity perhaps, are extremely informative in understanding the underlying drivers as well as processes and data streams that may improve the performance of future approaches. In the ocean, species richness was under-predicted (blue in Fig 1C) in several coral reef ecosystems (e.g., the Coral Triangle, Marianas Archipelago and Hawaiian Islands). On land, it was under-predicted in several montane forests (e.g., the Tropical Andes, Eastern Arc Mountains). These hyper-diverse ecosystems offer structurally-complex biogenic substrates [54, 55] that perhaps increase the available ecological niches and evolutionary pressures [8]. Regions where the model overpredicted biodiversity (red in Fig 1C) include steep biogeographic boundaries (e.g., the Andesite Line in Melanesia) and isolated islands (e.g., Madagascar, New Zealand, Hawaii and the Greater Antilles). These regions include biologically-isolated islands [7] and subphotic depths that lack biogenic and structurally complex seafloor habitat, like coral reefs and kelp forests. In addition, western ocean boundary regions were also poorer than predicted (arrows, Fig 1C). These regions are defined by the stable presence of major, high-velocity current systems (e.g., Agulhas, Kuroshio, Gulf Stream and Northern Brazilian) that facilitate the dispersal of waters and organisms to other regions [56, 57].

Latitudinal gradients have received significant attention in biogeography [2, 58], and as a result they appear in our analyses. However, even without latitude as an input, our models notably predicted the overall richness patterns well (Fig 1D). Were latitude important, then we would likely see some systematic pattern in Fig 1C and 1D. More likely is that latitude only provides a broad approximation of where there are favourable conditions for species richness, conditions now better captured by other variables. The northern-hemisphere bias in marine biodiversity in Fig 1D, for example, may not be a function of latitude, but rather the spatial overlap of favourable conditions such as depth, sunlight, and temperature [59]. To put this another way, latitude is not itself a mechanism, but a proxy that summarizes the confluence of individual variables across space. Now that those variables are themselves more available at

finer spatial resolutions and temporal scales, it makes sense to model the mechanisms themselves directly. A primary advantage of this approach is that while climate change will not affect latitude, many of the underlying variables associated with that latitude are moving rapidly [60]. So while the single variable of latitude may have helped historically in describing patterns of biodiversity, climate change is fundamentally altering what latitude itself has meant for ecology and evolution. Therefore, the usefulness of latitude itself in forecasting the impact of climate change is limited.

The data and model frameworks we used here importantly allowed us to quantify the contribution of environmental drivers in shaping biodiversity. Marine biodiversity was largely predicted by three variables while on land more variables are important (Fig 2A–2C). Sunlight and temperature were the most important, represented in 2 of the top 3 drivers in each domain (Fig 2A and 2B). In the ocean, depth was the dominant predictor as shallow marine regions may support both vibrant benthic and water column communities. Primary production exerted a stronger influence on terrestrial than marine species richness, seen in both the variable ranks (Fig 2A and 2B) and the functional forms (Fig 2D). This may be explained as primary producers on land are more constrained by light and temperature, while marine production is independently driven by nutrients and upwelling [21] in response to physical oceanographic processes. Though the functional form varies, biodiversity decreased with increasing vertical distance from sea-level, that is, at deeper depths and greater elevations (Fig 2D). Oxygen in the ocean and precipitation on land represented important, domain-specific biochemical constraints that influence species distributions [Fig 2A, 16, 17]. A full suite of pairwise relationships is presented in S4 Fig and S6 and S8 Figs present alternative comparisons and visualizations of this nature.

A strength of the ANN approach we used here is the ability to evaluate multiple drivers as well as their interactions across driver gradients. This may allow for a more representative understanding of any given location, where all the measured environmental variables coincide and interact to influence species distributions and yields more biologically relevant insights. We demonstrated this here in several shared as well as domain-specific drivers (Fig 3). Given the same depth or elevation, for example, temperature had roughly the opposite impact on terrestrial and marine biodiversity (Fig 3A). This was evident as the highest modelled species richness on land is mid-elevation locations with moderate temperatures. By contrast, in the ocean shallow and warm locations were the most biodiverse (Fig 3A). This same example also shows that the effect size of temperature on species richness is greater at both higher elevation and at shallower ocean depths. The intensity of the phenology of primary production had a strong effect on species richness across elevations. However, in the deep ocean biodiversity was low regardless of how seasons affect productivity (Fig 3B) or when considering the variability of dissolved oxygen (Fig 3C). Taken together, however, the relative impact of elevation on terrestrial biodiversity was negligible by comparison to the impact of depth on marine biodiversity. This was evident in the different scales of the *y* axes in the two columns of Fig 3A–3C as well as in the variable importance plots (Fig 2B and 2C).

Our analyses highlight how advances in data streams and models can bear dividends in predicting the present and future distribution of biodiversity. We provide a step toward a global model of biodiversity by focusing on the maintenance of species richness within a more robust analytical framework of environmental drivers [9]. Future attempts to move away from reductionist theory must reflect the pool of available resources considered in past hypotheses, appreciate the full empirical variability of the model drivers themselves, and assess the effect of driver interactions. How the model outputs deviate from the observation inputs identifies future areas in need of research development. Our ANN for marine biodiversity, for example, did not perform as well as that for the terrestrial domain (Fig 1C). Even though many data

sources have become more resolved in both space and time and more freely available [61], particularly for marine systems, there remains room for growth. Increased investments in biodiversity monitoring and earth observation systems are fundamental for advancing the analyses presented here. These investments should maintain and upgrade existing data streams and pipelines [e.g., remote sensing and biotelemetry, 62, 63], encourage promising emerging technologies [e.g., environmental DNA, 64, 65], and invest in autonomous and networked environmental sensing [66–68]. In particular, mechanistic and probabilistic models of species distributions will be improved by technological advances in biologging and tracking [63, 69, 70] that facilitate biodiversity monitoring and resolving the processes underlying species movements. Resolving the continuous variability of environmental drivers across ocean depths at a global scale will also improve the understanding of marine biodiversity.

Although a central goal of our approach here is to summarize data across terrestrial and marine domains, our results may lead to progress in applied contexts. Climate change is rapidly shifting ecosystems across the globe, impacting entire economic sectors such as fisheries and challenging their governance [12]. Ten of the top 15 terrestrial biodiversity drivers identified, and 6 of the top 15 marine biodiversity drivers, are directly mediated by climate (Fig 2A and 2B). Resolving how these drivers themselves will shift under different scenarios of climate change and applying this to forecasted models of future species distributions, may help us more fully grasp the risks climate change brings. Such forecasts will be improved by focusing on more direct drivers and deemphasizing distant surrogate variables such as latitude that mask underlying mechanisms. Lastly, conservation priorities based on the mechanistic approaches might focus on globally at-risk species [1, 71] as these groups have less adaptive capacity and are more prone to extinction.

## Supporting information

**S1 Table. All remotely-sensed variables are modeled from fullest temporal extent of available data streams, described in the main text methods.**
(DOCX)

**S1 Fig. Geographical distribution of model predictors and response.** All inputs are projected in a cylindrical equal-area projection centered on 195-degree latitude. Equatorial resolution is approximately 50 km × 50 km. Gray masks are where comparable metrics were used in separate domains, i.e. SST/SAT or $O_2$/$H_2O$. All PP metrics are prefixed with *ann* for interannual metrics and *sub* for intra-annual metrics, *AMP* prefixes refer to PP seasonal wavelet intensity followed by the 6 or 12-month period. *Twel_inten* and *six_*inten refers to the seasonal wavelet intensity of solar insolation. All biogeochemical constraints ($O_2$ and $H_2O$) are intra-annual metrics. Variability and tendency are measured as labeled in each panel: *mean*, *range*, *sd*, *CV*. See S1 Table for summary of metrics utilized in model training.
(DOCX)

**S2 Fig. Distributions of model features and response for terrestrial biodiversity model.** See full methods for description of development.
(DOCX)

**S3 Fig. Distributions of model features and response for the marine ANN biodiversity model.** See full methods for description of development.
(DOCX)

**S4 Fig. Full compilation of model input pairwise relationships with species richness.** Blue lines refer to marine domain and green terrestrial. The x-axis is the labeled 0–1 scaled

predictor, y-axis is always scaled ln(x+1) transformed 0–1 scaled species richness. Also see S9 Fig for raster visualization of all the features shown in this figure. See caption of S1 Fig for variable naming conventions.
(DOCX)

**S5 Fig. Example approximations of 6- and 12-month phenological wavelet power with morlet wavelet power spectrum.** Three examples of differing seasonal NDVI periodicities and the associated power spectrum. The continuous wavelet transformation applied to raster time series of Chl-A and NDVI is described in the methods and extensively in the annotated reference material.
(DOCX)

**S6 Fig. Ternary plots offer exploration beyond pairwise comparison of drivers.** These are ternary plots that show binned medians of species richness at varying row-scaled relative ratios of the labeled drivers to one another. These plots highlight the converged or divergent nature of global species richness relative to drivers. The gray areas are where particular combinations of drivers are not observed in the dataset. Ann refers to interannual PP metrics and sub refers to intra-annual metrics. All temperature metrics are summaries of intra-annual data. These figures are offered as an alternative way to visualize Fig 3 in the main text.
(DOCX)

**S7 Fig. Observed-versus-predicted for training and testing residuals.** The top row are terrestrial residuals, and the bottom are marine. On the left are the residuals of the full dataset with the color ramp utilized in the Fig 1C residual map. On the right are the residuals on the test set.
(DOCX)

**S8 Fig. 2D surface representations of the partial dependency plots from the main text.** This is an alternative visualization of Fig 3. Each x and y axis now represents the two predictors (x is always elevation/depth). To this plot we added a marginal rug of observed values on both axes to show the distribution of raw data across the full domain of observations.
(DOCX)

**S9 Fig. Alternative visualization of latitudinal gradient.** Top row is observed, bottom is predicted. Left column is terrestrial, right is marine. All points are single points of richness from the observed values and modeled predictions. These are the raw points by which the Fig 1D estimates median richness in latitudinal bins. The color ramp is the same used in Fig 1A and 1B.
(DOCX)

**S10 Fig. Observed global species richness without invertebrates.** Species richness where the marine data does not contain invertebrate taxa. It has been suggested that varied patterns between domains may emerge if marine richness data does not contain invertebrate data in the same way the terrestrial data does not currently. These are the outputs of the analysis presented in the manuscript but conducted on marine species richness data without invertebrate taxa. The broad scale trends presented in the manuscript appeared to show little difference to those observed here.
(DOCX)

**S11 Fig. Full compilation of model input pairwise relationships with species richness where marine richness does not contain invertebrate taxa.** Blue lines refer to marine domain without invertebrates and green terrestrial. The x-axis is the labeled 0–1 scaled predictor, y-

axis is always scaled ln(x+1) transformed 0–1 scaled species richness. See caption of S1 Fig for variable naming conventions.
(DOCX)

**S12 Fig. Partial dependency plots visualizing model input interactions with species richness, when marine invertebrates are not included in the marine ANN model.** These plots show the neural networks' approximation of driver richness relationships when two inputs co-vary–a key product and advantage of the ANN modelling framework. In this example, we see how species richness is expected to respond across both different temperatures and seasonality in primary productivity in response to changing depth and elevation. See main text for comparison to marine richness data with full suite of available taxa.
(DOCX)

**S13 Fig. Histograms of predicted versus observed richness across domains facetted by marine richness with or without invertebrates.** The hashed area represents the marine domain without invertebrate taxa. Here, (b) is identical to Fig 1D in the main text. The inclusion, or exclusion, of marine invertebrates, does not systematically alter the results of our analysis that compares modeled to observed richness in the marine ANN. As a note, neither of these issues affects the terrestrial analysis, which is separate, and for which no invertebrate data are available at the global scale.
(DOCX)

**S14 Fig. Map of residuals from the model that did not include marine invertebrates.** The terrestrial domain remains the same as the main text analysis, whereas the marine domain is showing residuals of predicting marine richness without invertebrate taxa. The observed pattern in Fig 1C, where there is an overlap of species-poor marine regions and high velocity boundary currents is retained in this analysis.
(DOCX)

**S15 Fig. Variable importance in marine domain without invertebrate taxa.** The top four most important variables remain the same as the model of the species richness data containing invertebrate taxa.
(DOCX)

**S16 Fig. Assessing the potential impact of sampling bias on the marine species distributions.** The above figure shows rarefaction or species accumulation curves for each 20˚ latitude bin for marine biodiversity when all species are included. Colors of 20˚ latitude bins correspond to rarefaction plot symbology below, where northern and southern hemispheres are split. This shows that sampling effort was not correlated with any spatial gradient of diversity. Thus, sampling did not significantly influence the observed biodiversity gradient depicted in Fig 1B and 1C.
(DOCX)

## Acknowledgments

A. Copenhaver provided logistical support for this project. S. Pimm provided helpful discussions and three anonymous reviewers improved earlier versions of this manuscript.

## Author Contributions

**Conceptualization:** Tyler O. Gagné, Clinton N. Jenkins, Steven J. Bograd, Elliott L. Hazen, Kyle S. Van Houtan.

**Data curation:** Tyler O. Gagné, Gabriel Reygondeau, Clinton N. Jenkins, Joseph O. Sexton, Kyle S. Van Houtan.

**Formal analysis:** Tyler O. Gagné, Gabriel Reygondeau.

**Funding acquisition:** Kyle S. Van Houtan.

**Investigation:** Tyler O. Gagné, Clinton N. Jenkins, Kyle S. Van Houtan.

**Methodology:** Tyler O. Gagné, Gabriel Reygondeau, Clinton N. Jenkins, Joseph O. Sexton, Steven J. Bograd, Elliott L. Hazen, Kyle S. Van Houtan.

**Project administration:** Tyler O. Gagné.

**Resources:** Kyle S. Van Houtan.

**Software:** Tyler O. Gagné.

**Supervision:** Tyler O. Gagné, Kyle S. Van Houtan.

**Validation:** Tyler O. Gagné.

**Visualization:** Tyler O. Gagné, Steven J. Bograd, Elliott L. Hazen, Kyle S. Van Houtan.

**Writing – original draft:** Tyler O. Gagné, Steven J. Bograd, Elliott L. Hazen, Kyle S. Van Houtan.

**Writing – review & editing:** Tyler O. Gagné, Gabriel Reygondeau, Clinton N. Jenkins, Joseph O. Sexton, Steven J. Bograd, Elliott L. Hazen, Kyle S. Van Houtan.

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
