## [Decision Letter · Decision Letter 0]

25 Jul 2019

PONE-D-19-15114

On the global drivers of biodiversity: unifying data, tools and domains.

PLOS ONE

Dear Mr. Gagne,

Thank you for submitting your manuscript to PLOS ONE. After careful consideration, we feel that it has merit but does not fully meet PLOS ONE’s publication criteria as it currently stands. Therefore, we invite you to submit a revised version of the manuscript that addresses the points raised during the review process.

I found this to be a really interesting paper that could be potentially far reaching. However, both reviewers have identified significant issues with the paper that need to be resolved. Reviewer 2 in particular has articulated numerous problems with the paper, including the data being used, and suggested the paper be rejected. Reviewer 1 felt major revisions could correct the problems. I am going to provide the authors with an opportunity to revise the manuscript as I think it could be a really useful contribution. However, if the required revisions to address the issues are not (or cannot be) made, the paper will have to be rejected.

My main issue with the paper is that it does not meet the PLoS standard for methods papers, which is spelled out in my editorial comments and also available online. That is, the paper does not make it clear why this method is better than any other method. There doesn't seem to be any value add of using this method; this is a requirement of a methods paper. Reviewer 2 has also raised a problem with the data being used and it is not clear how the authors have avoided the same problems of using such data as previous studies (or if they have). The methods section really needs some substantial expansion as much of what has been done is not clear to the reader. it should be very clear to the reader how this method is an improvement. 

Both reviewers have provided detailed comments to assist the authors in their review and I have provided editorial comments. The authors need to consider all the comments provided and must make it clear how they have addressed some rather serious issues with the manuscript. 

We would appreciate receiving your revised manuscript by Sep 08 2019 11:59PM. To enhance the reproducibility of your results, we recommend that if applicable you deposit your laboratory protocols in protocols.io, where a protocol can be assigned its own identifier (DOI) such that it can be cited independently in the future. For instructions see: http://journals.plos.org/plosone/s/submission-guidelines#loc-laboratory-protocols

We look forward to receiving your revised manuscript.

Kind regards,

Heather M. Patterson, Ph.D.

Academic Editor

PLOS ONE

Journal Requirements:

1. We note that you have stated that you will provide repository information for your data at acceptance. Should your manuscript be accepted for publication, we will hold it until you provide the relevant accession numbers or DOIs necessary to access your data. If you wish to make changes to your Data Availability statement, please describe these changes in your cover letter and we will update your Data Availability statement to reflect the information you provide.

Reviewers' comments:

Reviewer's Responses to Questions

**Comments to the Author**

1. Is the manuscript technically sound, and do the data support the conclusions?

Reviewer #1: Yes

Reviewer #2: No

2. Has the statistical analysis been performed appropriately and rigorously? 

Reviewer #1: I Don't Know

Reviewer #2: No

3. Have the authors made all data underlying the findings in their manuscript fully available?

Reviewer #1: Yes

Reviewer #2: Yes

4. Is the manuscript presented in an intelligible fashion and written in standard English?

Reviewer #1: Yes

Reviewer #2: Yes

5. Review Comments to the Author

Reviewer #1: This manuscript is an important contribution to the study of global patterns of biodiversity. I think that it is exciting to see an example of using a neural network approach to analyse questions of biodiversity. As the authors indicate, a neural network approach is one way to integrate the wealth of global data for drivers and biodiversity that we have collected. The novelty of this approach is exciting and I have been waiting to see such an application of a neural net for this question. However, I believe that this manuscript requires more development before publication. Given the novelty of the neural network approach, there are several concerns that need to be addressed.

First the authors demonstrate the benefit of using a NN approach to integration of multiple types of drivers of biodiversity patterns, and in particular, I believe that the authors do demonstrate that a NN approach can improve understanding of driver interactions. However, I think that they do not clearly or strongly present why or how a neural net approach can be better than using more traditional analyses like mixed effects models or GAMs (e.g., L120-122 is not strongly supported by their data or discussion). Additionally, despite my excitement to see the use of ANNs for this research question, I have concerns about the generality of the method. As driver data are added or removed, does this change the results? How sensitive are the results to differences in data processing (e.g., scaling)? In lines 255-260 the authors mention that ANNs can be used to parameterize interactions between variables without having to be explicitly defined, is this ever a draw back?

Second, there needs to be a more developed explanation of what an ANN is and how it works (e.g., the acronym is only defined fully “artificial neural network” in the Fig 1a caption) such that a general audience can better assess the results. I think more development of this explanation is also important because the authors identify ANNs as a way forward for studying patterns of biodiversity. In addition to this, I think that more detail needs to be included about data processing, for example, the species richness observations were scaled, but I did not see how the data were scaled, and whether there is a chance of this affecting the model output or interpretation.

Lastly, I am surprised by the lack of discussion of the latitudinal gradient hypothesis. I recognize that the authors are trying to make the point that analysis of direct drivers rather than proxies/surrogates is an advantage, but believe that readers would benefit from more discussion of this in the context of the existing literature. Furthermore, I think that some of the implications or interpretations of their main findings about patterns of species richness would benefit from more explanation. Figure 1C leaves much to be discussed. The authors hint at why the model output differs and highlight certain areas, but I think this requires more discussion. Without a scale, it is difficult to tell how substantial the residual errors are. But this is also a place to add more discussion and connection with the existing literature. Are there driver datasets that the authors believe are missing?

Minor/line specific comments/examples:

L 34: I find ‘geographic domains’ unclear, is this meant to terrestrial/marine

L 37: I think this sentence can be unpacked. The functional form of what? This is not immediately clear unless you have read the manuscript.

L 59: Why not include marine invertebrates in the main text results? They do not appear to change the results substantially and ultimately it looks like terrestrial and marine biodiversity are compared and discussed separately. Furthermore, I think this leads to the question, of why seek to unify the marine and terrestrial patterns of biodiversity? What is the benefit of attempting to unite these systems when you identify such differences. I think that the unification of domains (marine and terrestrial) is an overstatement and actually takes away from the take home message of the paper. Especially since the data inputs and models for marine and terrestrial systems were performed separately. Or now that I am rereading it in L224, were the two discrete feed forward neural networks not for marine and terrestrial biodiversity separately? The comparison of drivers, their relative importance, interactions, and analogous drivers is insightful, but the framing of a unification of domains is unnecessary.

L 63: I think it is worth including the primary producers in at least a separate analysis because this is a substantial amount of diversity to be excluding from the analysis of global richness patterns. Especially given that you discuss the potential importance of structurally complex biogenic substrates (L83). Would this result have changed if diversity of the primary producers had been included? What is the collinearity between primary productivity measures (NDVI and Chl-A) and diversity of the primary producers?

L67: What new information is provided about the driver relationships? Can these not be considered using traditional methods of analysis? I think this needs more discussion.

L76: What is the input-output relationship?

L86: Why do you think that depth did not capture the overpredicted regions in oceanic habitats too deep for major coral reefs? This may be a place where it also worth discussing the limitations in the driver data. What datasets are only available for surface waters and how does this affect the model’s ability to predict diversity patterns? (e.g., statements about the effect of O2 on richness should consider that this is only data down to 100 m – or only 30 m?, does this match the species distribution data? And how would this affect the model results, particularly Fig 2c?)

L88: Why would high-velocity current systems be more species-poor than expected?

L91: What are the favourable conditions for species richness, those drivers which you identified in the previous paragraph?

L93: What data do you show this pattern of habitable depths and temperatures? You jump to metabolic constraints without explaining this.

L107: How does the ANN deal with collinearity? This statement about patterns in spatial patterns of global fisheries production makes it seem like depth might be a proxy for a suite of other drivers. I am also unsure of the purpose of this comparison.

L133: Synthesizing data across what kind of domain?

L148: In what ways do the underlying datasets ultimately differ?

L151: What was the resolution of these datasets?

L177: Provide reference to supplementary information

L229: The approaches used to improve ANN model interpretation need to be discussed more. What are perturbation and resampling and how do these help us understand driver effects on richness?

L261 - 277: I found this discussion helpful.

L345: Capitalize ‘science’ – and a general check over references, this always gets me too.

Figure 1c: labelling the +, - on the scale could be replaced with over/under prediction to make it easier for the reader.

Figure 2a: Are the predictors intensity/concentration/level of these drivers or are they variance?

Figure 3c: Where do you discuss this in the text? Are these results supported by the literature? It also does not make sense to look at the model effect of O2 variance in the deep ocean if the data for these depths were not used. I would expect that concentration of O2 would be more important than variance, even though the relative importance analysis suggests that oxygen CV explains patterns of richness better.

Supplementary info: It would be helpful to reference specific figures in the supplement.

Table S1: It would be helpful to have the spatial and temporal resolution of the datasets included here.

Figure S10: Is this the expected richness from the model?

Reviewer #2: General comment

Gagne et al. present an exciting prospect of training emerging AI/ML techniques into global biodiversity assessment. They should be commended in their cleaning of global biodiversity informatics database as this is no small task. I was excited to read this paper as these upcoming techniques show great promise in the era of big data and I support/commend the authors push back against the 'black box' dismissal of these powerful techniques. However, I have some serious reservations about the appropriateness in how these data have been used and the inference being drawn from them. It appears that the authors have not properly addressed some of the major issues in using sparse point observation data for community level analyses. Overall the claims of new insight coming from these analytics are not shown in this paper. Rather it seems that the authors have revisited analyses that have been undertaken many times before (at least in the terrestrial space) using a relatively underused method. They have not shown any value-add to using these new methods.

Major comments

I have serious reservations about the appropriateness of the species point observation data being used in the marine database. The authors have done an impressive job of name matching and filtering these data, however, I see nothing in the description of the methods to suggest that any work has been done to ensure the aggregated richness values are representative of true species richness rather than a combination of richness and sampling bias. This has the potential to seriously bias inference when using these kinds of data and has to be addressed. Indeed, they state that their database covers ~17% of marine species suggesting significant gaps in their aggregated richness data. As far as I’m away ANN are not able to counter these kinds of biases (yet), so the authors will need to make a strong argument as to why their data are fit-for-purpose.

Following from that, I question why, if point observation data are appropriate for the marine models, are they are not used in the terrestrial models? GBIF contains 100s of thousands of terrestrial species that are not birds, mammals and amphibians (which only represent a tiny fraction of terrestrial diversity). The authors should be using the same kinds of primary data if they are to make inference between the two.

The authors argue that their analyses ‘highlight how advances in data streams and models will bear dividends in predicting the present and future distribution of biodiversity”. Beyond the use of neural networks to undertake the analysis, I don’t see many advances in this paper. In fact, the authors are using a very coarse spatial resolution data when compared with modern global biodiversity assessments. It would be more impressive to see this analysis carried out a spatial resolution more appropriate to the processes that they are attempting to describe BUT see my comment on the appropriateness of the baseline biological data being used.

While the authors argue that a strength of their analytics is the assessment of multiple drivers and their interactions, I fail to see how this is any different to completing this study using a different analytical technique (multiple regression analysis for instance). The paper needs to establish more clearly, where the actual benefits of ANN methods lies over a more common approach. I am a big supporter of modern AI/ML techniques and I think this paper sells them a little short – instead relying on the current excitement for the ‘machine learning magic sauce’. For instance, do you arrive at any massively different inference using a different analytical technique? If so, why is this? Why is inference from ANN more appropriate?

One of the big advantages of AI/ML techniques is their ability to integrate across VERY large datasets to draw inference. I am not too sure the data being used by the authors could be classed as this – they make no mention of how many 50km pixels were used in the analysis – which brings be back to my point above about what are the advantages of using ANN?

The methods need to be significantly expanded to improve clarity. In particular, the structuring of the model and the assumptions being made about the inputs. Maybe some of my misgivings about this study would be shown to be misplaced with a more detailed description of the methods.

Other comments

Introduction paragraph 1: But your data rely on IUCN range maps, the publically available data do not even cover all vertebrates. Vertebrates are a small fraction of biological diversity as a whole and there are 100s of papers that focus on these taxa using the same baseline data that you use.

Line 52-53: Rephrase.

Line 66: Define ANN

Line 98-100: While this may be true. It’s also true that these are the areas most easily studied and your results represent a potential sampling bias in the underlying datasets.

Figure S13: Your terrestrial results with/without marine invertebrates show a slightly different response? Are these both modelled within the same network? If so, you need to explain more clearly why this is appropriate, particularly given your very different data being used for target and feature variables.

6. PLOS authors have the option to publish the peer review history of their article (what does this mean?). If published, this will include your full peer review and any attached files.

Reviewer #1: No

Reviewer #2: No

---

## [Author Response · Author response to Decision Letter 0]

8 Nov 2019

Please see the 'response to reviewers' document, where we carefully address and respond to all reviewer comments.

---

## [Decision Letter · Decision Letter 1]

31 Dec 2019

PONE-D-19-15114R1

Towards a global understanding of the drivers of marine and terrestrial biodiversity

PLOS ONE

Dear Dr. Van Houtan,

Thank you for submitting your manuscript to PLOS ONE. After careful consideration, we feel that it has merit but does not fully meet PLOS ONE’s publication criteria as it currently stands. Therefore, we invite you to submit a revised version of the manuscript that addresses the points raised during the review process.

I think the manuscript is greatly improved so I thank the authors for their efforts. Unfortunately, there is still disagreement between the reviewers, with Reviewer 1 recommending the paper be accepted and Reviewer 2 recommending rejection. While I understand the points made by Reviewer 2, I am inclined to agree with Reviewer 1 and have provided just a few minor corrections and clarifications to be addressed. Reviewer 2 has also noted that Figure 1 does not cover the full extent of the models and should be expanded. 

We would appreciate receiving your revised manuscript by Feb 14 2020 11:59PM. To enhance the reproducibility of your results, we recommend that if applicable you deposit your laboratory protocols in protocols.io, where a protocol can be assigned its own identifier (DOI) such that it can be cited independently in the future. For instructions see: http://journals.plos.org/plosone/s/submission-guidelines#loc-laboratory-protocols

We look forward to receiving your revised manuscript.

Kind regards,

Heather M. Patterson, Ph.D.

Academic Editor

PLOS ONE

Reviewers' comments:

Reviewer's Responses to Questions

**Comments to the Author**

1. If the authors have adequately addressed your comments raised in a previous round of review and you feel that this manuscript is now acceptable for publication, you may indicate that here to bypass the “Comments to the Author” section, enter your conflict of interest statement in the “Confidential to Editor” section, and submit your "Accept" recommendation.

Reviewer #1: All comments have been addressed

Reviewer #2: (No Response)

2. Is the manuscript technically sound, and do the data support the conclusions?

Reviewer #1: Yes

Reviewer #2: No

3. Has the statistical analysis been performed appropriately and rigorously? 

Reviewer #1: Yes

Reviewer #2: No

4. Have the authors made all data underlying the findings in their manuscript fully available?

Reviewer #1: Yes

Reviewer #2: Yes

5. Is the manuscript presented in an intelligible fashion and written in standard English?

Reviewer #1: Yes

Reviewer #2: Yes

6. Review Comments to the Author

Reviewer #1: The authors have made substantial revisions to this manuscript and have addressed my comments sufficiently. I believe that this manuscript has been greatly improved and does a much better job of demonstrating the use of ANNs in ecology and deeper insight into some of the processes that can affect global patterns of species richness. The revised title reflects how the authors have reframed the purpose of this manuscript and this article is much easier for a reader to follow. The description of the methods and datasets used is more straightforward, allowing the reader to better understand and assess the method and potential limitations. The authors have also improved the conclusions and interpretation of the results.

Reviewer #2: I continue to commend the authors on the significant amount of work undertaken to name match species records and remove erroneous species observations from a set of global biodiversity informatics databases. I also thank the authors for their clarifications of the methods used in their study. However, I still have major concerns with their use of point observation data to model species richness and the inference being drawn from these models. I do not think that the changes made to the manuscript are sufficient to make their use of these data appropriate. The potential problems with developing models of richness from these kinds of data are well established (see below for some references) and the authors have not provided any bias corrections to the aggregate richness measure or shown why such corrections should not be needed.

The authors have not shown that their estimates of marine richness are valid approximations of the unobserved true richness – at the spatial scale that the models are fitted. While the rarefaction curves provided by the authors in this revised version show no latitudinal sampling bias when the data are aggregated to 20 degree bands (25,000 times more coarse than the resolution used in the modelling exercise), they do not show that biases do not exist at the scale being modelled or that other kinds of spatial bias do not exist. They do not show that the spatial patterns the ANN is fitting to approximate the true patterns in species richness. I would guess that this is not the case and that models are being fitted to pixels with a high degree of under sampling which include spatial variation that does not represent the true variation in species richness. It is highly likely that between pixels there are large differences in the adequacy of sampling effort, that there are taxonomic biases (i.e. do all pixels contain records from all taxonomic groups studied?), and the ecological community assembled from these point data are not representative of the expected community in these pixels. If ANN are capable of correcting these kinds of biases, the authors have not shown this.

If we believe the author’s argument that point observation data are appropriate for use in the marine realm to model species richness via ANN then we have to disagree with the assertion that the authors are using the best available data for both the marine and terrestrial realms. Why are the point observation data in the terrestrial realm less appropriate to use? There are >400k terrestrial species in GBIF, far more than the 23k vertebrate range maps used in this study. I also note that the global catalogue of reptile range maps is now publicly available, as such even using range maps, the data used here are not the best available.

If it is the intent of the authors to highlight methodological advantages of ANN over more traditional methods, then they still have not shown that. The short textual description which focuses on the theoretical advantages of such methods – over statistical regression techniques – is a good start but they should show some comparison of the improvements in prediction and inference that can be made from such methods on the kinds of data being modeled. Have the authors attempted to build a regression model from these data to enable a comparison?

If the intent is to highlight the advantages of using ANN over traditional methods, then perhaps the easiest path forward would be to drop the marine component of these analyses, focus on drawing inference from the terrestrial range maps, and include some contrast with other established methods.

Fig 1: This map only shows spatial results for approx.. 45 S to 45 N but the models were fitted to areas further north and south (1d), these maps need to be expanded.

Some references relating to the difficulties with modelling species richness from point observation databases are below.

Geographical sampling bias in a large distributional database and its effects on species richness-environment models. https://onlinelibrary.wiley.com/doi/full/10.1111/jbi.12108

Mapping the biodiversity of tropical insects: species richness and inventory completeness of African sphingid moths. https://onlinelibrary.wiley.com/doi/full/10.1111/geb.12039

Spatial bias in the GBIF database and its effect on modelling species’ geographic distributions. https://www.sciencedirect.com/science/article/pii/S1574954113001155

Strengths and weaknesses of museum and national survey datasets for predicting regional species richness: comparative and combined approaches. https://onlinelibrary.wiley.com/doi/full/10.1111/j.1366-9516.2005.00164.x

A comparison of methods for mapping species ranges and richness. https://onlinelibrary.wiley.com/doi/10.1111/j.1466-8238.2006.00257.x

7. PLOS authors have the option to publish the peer review history of their article (what does this mean?). If published, this will include your full peer review and any attached files.

Reviewer #1: No

Reviewer #2: No

---

## [Author Response · Author response to Decision Letter 1]

6 Jan 2020

please see attached document. All requested changes were made :)

---

## [Editor Report · Decision Letter 2]

8 Jan 2020

Towards a global understanding of the drivers of marine and terrestrial biodiversity

PONE-D-19-15114R2

Dear Dr. Van Houtan,

We are pleased to inform you that your manuscript has been judged scientifically suitable for publication and will be formally accepted for publication once it complies with all outstanding technical requirements.

With kind regards,

Heather M. Patterson, Ph.D.

Academic Editor

PLOS ONE
---

## [Editor Report · Acceptance letter]

13 Jan 2020

PONE-D-19-15114R2 

Towards a global understanding of the drivers of marine and terrestrial biodiversity 

Dear Dr. Van Houtan:

I am pleased to inform you that your manuscript has been deemed suitable for publication in PLOS ONE. Congratulations! Your manuscript is now with our production department. 

With kind regards,

on behalf of

Dr. Heather M. Patterson 

Academic Editor

PLOS ONE